# DETECTING OUT-OF-DISTRIBUTION THROUGH THE LENS OF NEURAL COLLAPSE

## ABSTRACT

Out-of-distribution (OOD) detection is essential for the safe deployment of AI. Particularly, OOD detectors should generalize effectively across diverse scenarios. To improve upon the generalizability of existing OOD detectors, we introduce a highly versatile OOD detector, called Neural Collapse inspired OOD detector (NC-OOD). We extend the prevalent observation that in-distribution (ID) features tend to form clusters, whereas OOD features are far away. Particularly, based on the recent observation, Neural Collapse, we further demonstrate that ID features tend to cluster in proximity to weight vectors. From our extended observation, we propose to detect OOD based on feature proximity to weight vectors. To further rule out OOD samples, we leverage the observation that OOD features tend to reside closer to the origin than ID features. Extensive experiments show that our approach enhances the generalizability of existing work and can consistently achieve state-of-the-art OOD detection performance across a wide range of OOD Benchmarks over different classification tasks, training losses, and model architectures.

## 1 INTRODUCTION

Machine learning models deployed in practice will inevitably encounter samples that deviate from the training distribution. As a classifier cannot make meaningful predictions on test samples that belong to unseen classes during training, it is important to actively detect and handle Out-of-Distribution (OOD) samples. Considering the diverse application scenarios, an effective OOD detector should generalize across classification tasks of different input resolutions, number of classes, classification accuracy, as well as classifiers under different training schemes and architectures.

Since Nguyen et al. (2015) reveals that neural networks tend to be over-confident on OOD samples, an extensive body of research has been focused on developing effective OOD detection algorithms. One line of work designs OOD scores over model output space (Liang et al., 2018; Liu et al., 2020; Hendrycks et al., 2019; Sun et al., 2021; Sun & Li, 2022). Another line of work focuses on the feature space, where OOD samples are observed to deviate from the clusters of ID samples (Tack et al., 2020; Lee et al., 2018; Sun et al., 2022) and builds an auxiliary model for OOD detection. Specifically, Lee et al. (2018) detects OOD based on the Mahalanobis distance (Mahalanobis, 2018) between the feature and the mixture of Gaussian distribution learned from training features; Sun et al. (2022) measures OOD-ness based on the k-th nearest neighbor distance to training features. While previous efforts have significantly improved OOD detection performance, we observe in Table 1 that existing work does not concurrently achieve state-of-the-art performance across different classification tasks, as competitive approaches on ImageNet(Deng et al., 2009) OOD benchmarks perform suboptimally on CIFAR-10(Krizhevsky et al., 2009) OOD benchmarks, and vice versa.

In this work, we aim to improve upon the generalizability and build a versatile OOD detector across diverse scenarios. To this end, we start with the prevalent observation discovered in Tack et al. (2020); Lee et al. (2018); Sun et al. (2022) that ID features tend to form clusters at the penultimate layer, i.e., the layer before the linear classification head, whereas OOD features reside far away, as shown in Figure 1 *Left*. To better understand the phenomenon, we ask:

*Where do ID features tend to form clusters?*

To answer this question, we draw insights from the recent finding *Neural Collapse* (Papyan et al., 2020), which describes the limiting behavior of features at the penultimate layer and the linear

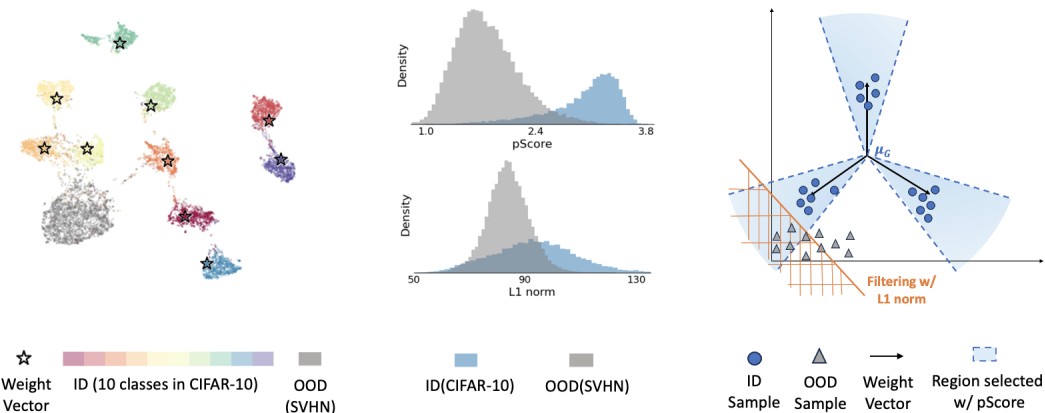

Figure 1: **Illustration of our framework inspired by Neural Collapse.** *Left:* On the penultimate layer, features of ID samples reside near the weight vectors of the linear classification head, visualized with UMAP(McInnes et al., 2018). *Middle:* ID and OOD samples can be distinguished by our pScore defined in Section 3, which measures feature proximity to weight vectors. Also, OOD samples tend to reside closer to the origin than ID samples, illustrated with L1 norm. *Right:* Thresholding on pScore selects infinite hypercones centered at weight vectors, shaded in blue. And the L1 norm in NCScore helps filter out OOD samples near the origin that fall within the hyper-cones. Note that *Left* and *Middle* present experimental results on a CIFAR-10 ResNet-10 classifier with OOD set SVHN. *Right* depicts the idea of our scheme in two dimensions on a three-class classifier.

classification head when the training epoch goes to infinity. We demonstrate that as a deterministic effect of Neural Collapse, features of training samples will converge towards the weight vectors of the linear classification head. Since ID test samples are drawn from the same distribution as training samples, we hypothesize features of ID test samples to reside close to weight vectors as well. This is demonstrated in Figure 1 *Left*, where ID features form clusters near weight vectors while OOD features reside far away. Based on this, we propose to detect OOD based on feature proximity to weight vectors and we define a proximity score as the norm of the projection of weight vectors onto the sample feature. In comparison to prior work (Lee et al., 2018; Sun et al., 2022) which builds an auxiliary model from training features, our alternative characterization of ID clustering is more computationally efficient and brings in a performance gain.

Despite the effectiveness of our proximity score in detecting OOD samples, supported by Figure 1 *Middle Upper*, some OOD features can still fall within the highlighted hypercones centered at weight vectors and cannot be distinguished from ID features under our proximity score, as illustrated in Figure 1 *Right*. To further filter out such OOD samples, we adopt a complementary perspective and utilize the observation that OOD features tend to reside closer to the origin (Tack et al., 2020; Sun et al., 2022). Specifically, we add the L1 norm of the feature to the proximity score to filter out OOD near the origin. Thresholding on the score, we have **N**eural **C**ollapse inspired **OOD** detector (NC − OOD): A lower score indicates a higher chance of OOD.

With extensive experiments, we show that our detector consistently achieves or improves state-of-the-art performance across OOD benchmarks on different classification tasks (ImageNet, CIFAR-10 & CIFAR-100), across different training objectives (cross-entropy & contrastive loss), and across different network architectures (ResNet & DenseNet). This improves the generalizability of existing methods. We further conduct a comprehensive ablation study to understand the effectiveness of each component in our OOD detector. We summarize our main contributions below:

- **Understanding ID feature clustering through the lens of Neural Collapse:** We delve into the prevalent observation that ID features tend to form clusters whereas OOD features reside far away. Based on Neural Collapse, we further demonstrate that features of training samples will converge towards weight vectors of the last layer. Taking one step further, we hypothesize and demonstrate that ID test samples tend to reside close to weight vectors whereas OOD samples are far away.

- **OOD detection methods:** Based on our understanding, we propose to detect OOD samples based on feature proximity to weight vectors. We further rule out OOD samples using the observation that OOD features tend to be closer to the origin. Our proposed OOD detector has remarkable computational efficiency and rapid inference speed.

Table 1: **Our proposed OOD detector achieves high AUROC across CIFAR-10 and ImageNet OOD benchmark.** AUROC score is reported for OOD detection across 6 different CIFAR-10 OOD benchmarks and 4 ImageNet OOD benchmarks. The larger the value, the better. The best result is highlighted in **bold** and the second best result is underlined.

| | Method | CIFAR-10 OOD Benchmark | | | | | | | ImageNet OOD Benchmark | | | | |
|---|---|---|---|---|---|---|---|---|---|---|---|---|---|
| | | SVHN | LSUN-C | LSUN-R | iSUN | Place365 | Texture | AVG | iNaturalist | SUN | Places | Texture | Avg |
| | MSP | 91.29 | 93.80 | 92.73 | 92.12 | 88.63 | 88.50 | 91.18 | 87.74 | 80.63 | 79.76 | 79.61 | 81.99 |
| CIFAR-10 Strong | ODIN | 89.12 | 98.39 | 97.78 | 97.37 | **91.85** | 89.41 | 93.99 | 89.66 | 84.59 | 81.78 | 85.62 | 85.41 |
| | Energy | 91.32 | 98.05 | 96.14 | 95.59 | 91.03 | 89.37 | 93.59 | 89.95 | 85.89 | 82.86 | 85.99 | 86.17 |
| | Maha | 95.67 | 92.09 | **97.87** | **97.93** | 68.44 | 85.90 | 89.65 | 52.65 | 42.41 | 41.79 | 85.01 | 55.17 |
| | KNN | 95.52 | 96.84 | 96.06 | 95.52 | 90.85 | 94.71 | 94.92 | 86.47 | 80.72 | 75.76 | 97.07 | 85.01 |
| | fDBD | 91.98 | 97.18 | 95.53 | 94.97 | 90.39 | 89.64 | 93.28 | 91.14 | 84.81 | 82.54 | 86.18 | 86.17 |
| ImageNet Strong | GradNorm | 79.85 | 96.21 | 90.67 | 89.34 | 82.71 | 81.72 | 86.75 | 93.86 | 90.16 | 86.14 | 90.66 | 90.21 |
| | ReAct | 92.44 | 95.96 | 96.86 | 96.38 | 90.90 | 91.30 | 93.97 | 96.22 | **94.20** | **91.58** | 89.80 | 92.95 |
| | DICE | 89.73 | **99,21** | 95.42 | 94.87 | 89.77 | 88.57 | 92.93 | 94.49 | 90.83 | 87.48 | 90.30 | 90.77 |
| | DICE + ReAct | 91.54 | 98.11 | 96.89 | 96.27 | 87.72 | 91.73 | 93.71 | 96.24 | 93.94 | 90.67 | 92.74 | 93.40 |
| | NC-OOD | **96.17** | 98.15 | 96.58 | 96.04 | 90.48 | **94.75** | **95.36** | **96.94** | 93.46 | 89.91 | **95.65** | **94.15** |

- **Experimental Analysis:** Through extensive experiments, we show our proposed OOD detector achieves or improves state-of-the-art OOD detection performance across a wide range of OOD benchmarks. We also conduct a comprehensive ablation study to understand the effectiveness of each component in our detector.

## 2 Problem Setting

We consider a data space $\mathcal{X}$, a class set $\mathcal{C}$, and a classifier $f : \mathcal{X} \to \mathcal{C}$, which is trained on samples *i.i.d.* drawn from joint distribution $\mathbb{P}_{\mathcal{X}\mathcal{C}}$. We denote the marginal distribution of $\mathbb{P}_{\mathcal{X}\mathcal{C}}$ on $\mathcal{X}$ as $\mathbb{P}^{in}$. And we refer to samples drawn from $\mathbb{P}^{in}$ as In-Distribution (ID) samples. In the real world, the classifier $f$ may encounter $\boldsymbol{x} \in \mathcal{X}$ which is not drawn from $\mathbb{P}^{in}$. We refer to such samples as Out-of-Distribution (OOD) samples.

Since a classifier cannot make meaningful predictions on OOD samples from classes unseen during training, it is important to distinguish between In-Distribution (ID) samples and Out-of-Distribution (OOD) samples for the reliable deployment of machine learning models. Moreover, considering the diverse application scenarios, an ideal OOD detection scheme should generalize across classification tasks of different input resolutions, number of classes, etc. In this work, we study the previous observation that ID features tend to form clusters. We draw insights from Neural Collapse to understand the phenomenon and we propose an OOD detector based on our understanding.

## 3 OOD Detection through the Lens of Neural Collapse

In this section, we propose an OOD detector drawing inspiration from Neural Collapse. Specifically, we dive into the observation in Lee et al. (2018); Sun et al. (2022) showing that ID features tend to form clusters and OOD features deviate from the ID clusters. To better understand the phenomenon, we ask the question:

*Where do ID features tend to form clusters?*

Leveraging insights from Neural Collapse, we show that features of training examples will converge towards weight vectors. We therefore expect features of ID samples, which are drawn from the same distribution as in training samples, to reside close to weight vectors as well whereas OOD samples reside far away, as shown in Figure 1 *Right*, We therefore propose to detect OOD based on feature proximity to weight vectors. We further filter OOD with the additional observation that OOD features tend to reside closer to the origin and we propose our Neural Collapse inspired OOD detector. We illustrate and summarize our scheme in Figure 1.

### 3.1 NEURAL COLLAPSE: CONVERGENCE OF FEATURES TOWARDS WEIGHT VECTORS

We now establish from Neural Collapse the relationship between penultimate layer features and the weight vectors of the linear projection head.

Neural Collapse, first observed in Papyan et al. (2020), occurs on the penultimate layer across canonical classification settings. To formally introduce the concept, we use $\boldsymbol{h}_{i,c}$ to denote the penultimate layer feature of the $i_{th}$ training sample with ground truth label $c$. And Neural Collapse is framed in relation to

- the feature global mean, $\boldsymbol{\mu}_G = \text{Ave}_{i,c}\boldsymbol{h}_{i,c}$, where Ave is the average operation;
- the feature class means, $\boldsymbol{\mu}_c = \text{Ave}_i\boldsymbol{h}_{i,c}, \ \forall c \in \mathcal{C}$;
- the within-class covariance, $\boldsymbol{\Sigma}_W = \text{Ave}_{i,c}(\boldsymbol{h}_{i,c} - \boldsymbol{\mu}_c)(\boldsymbol{h}_{i,c} - \boldsymbol{\mu}_c)^T$;
- the linear classification head, i.e. the last layer of the NN, $\arg\max_{c \in \mathcal{C}} \boldsymbol{w}_c^T\boldsymbol{h} + b_c$, where $\boldsymbol{w}_c$ and $b_c$ are parameters corresponding to class $c$.

Overall, Neural Collapse comprises four inter-related limiting behaviors in the Terminal Phase of Training (TPT) where training error vanishes and the training loss is trained towards zero:

**(NC1) Within-class variability collapse:** $\boldsymbol{\Sigma}_W \to \boldsymbol{0}$

**(NC2) Convergence to simplex equiangular tight frame (ETF):**
$$|\|\boldsymbol{\mu}_c - \boldsymbol{\mu}_G\|_2 - \|\boldsymbol{\mu}_{c'} - \boldsymbol{\mu}_G\|_2| \to 0, \ \forall \ c, \ c'$$
$$\frac{(\boldsymbol{\mu}_c - \boldsymbol{\mu}_G)^T(\boldsymbol{\mu}_{c'} - \boldsymbol{\mu}_G)}{\|\boldsymbol{\mu}_c - \boldsymbol{\mu}_G\|_2\|\boldsymbol{\mu}_{c'} - \boldsymbol{\mu}_G\|_2} \to \frac{|\mathcal{C}|}{|\mathcal{C}| - 1}\delta_{c,c'} - \frac{1}{|\mathcal{C}| - 1}$$

**(NC3) Convergence to self-duality:** $\dfrac{\boldsymbol{w}_c}{\|\boldsymbol{w}_c\|_2} - \dfrac{\boldsymbol{\mu}_c - \boldsymbol{\mu}_G}{\|\boldsymbol{\mu}_c - \boldsymbol{\mu}_G\|_2} \to \boldsymbol{0}$

**(NC4) Simplification to nearest class center:** $\arg\max_{c \in \mathcal{C}} \boldsymbol{w}_c^T\boldsymbol{h} + b_c \to arg\min_{c \in \mathcal{C}} \|\boldsymbol{h} - \boldsymbol{\mu}_c\|_2$

In the following, we derive from **(NC1)** and **(NC3)** and show that training ID features will converge to weight vectors of the linear classification head up to scaling.

**Theorem 1.** *(NC1) and (NC3) imply that for any $i$ and $c$, we have*
$$(\boldsymbol{h}_{i,c} - \boldsymbol{\mu}_G) \to \lambda\boldsymbol{w}_c \tag{1}$$
*in the Terminal Phase of Training, where $\lambda = \dfrac{\|\boldsymbol{\mu}_c - \boldsymbol{\mu}_G\|_2}{\|\boldsymbol{w}_c\|_2}$.*

*Proof.* Considering that matrix $(\boldsymbol{h}_{i,c} - \boldsymbol{\mu}_c)(\boldsymbol{h}_{i,c} - \boldsymbol{\mu}_c)^T$ is positive semi-definite for any $i$ and $c$. $\boldsymbol{\Sigma}_W \to \boldsymbol{0}$ therefore implies $(\boldsymbol{h}_{i,c} - \boldsymbol{\mu}_c)(\boldsymbol{h}_{i,c} - \boldsymbol{\mu}_c)^T \to \boldsymbol{0}$ and $\boldsymbol{h}_{i,c} - \boldsymbol{\mu}_c \to \boldsymbol{0}, \ \forall i, c$. With algebraic manipulations, we then have
$$\frac{\boldsymbol{h}_{i,c} - \boldsymbol{\mu}_G}{\|\boldsymbol{\mu}_c - \boldsymbol{\mu}_G\|_2} - \frac{\boldsymbol{\mu}_c - \boldsymbol{\mu}_G}{\|\boldsymbol{\mu}_c - \boldsymbol{\mu}_G\|_2} \to \boldsymbol{0}, \ \forall i, c \tag{2}$$
Applying the triangle inequality, we have
$$|\frac{\boldsymbol{h}_{i,c} - \boldsymbol{\mu}_G}{\|\boldsymbol{\mu}_c - \boldsymbol{\mu}_G\|_2} - \frac{\boldsymbol{w}_c}{\|\boldsymbol{w}_c\|_2}| \leq |\frac{\boldsymbol{h}_{i,c} - \boldsymbol{\mu}_G}{\|\boldsymbol{\mu}_c - \boldsymbol{\mu}_G\|_2} - \frac{\boldsymbol{\mu}_c - \boldsymbol{\mu}_G}{\|\boldsymbol{\mu}_c - \boldsymbol{\mu}_G\|_2}| + |\frac{\boldsymbol{w}_c}{\|\boldsymbol{w}_c\|_2} - \frac{\boldsymbol{\mu}_c - \boldsymbol{\mu}_G}{\|\boldsymbol{\mu}_c - \boldsymbol{\mu}_G\|_2}|. \tag{3}$$
Given both terms on RHS converge to $\boldsymbol{0}$ and shown by **(NC3)** and equation 2, the LHS also converges to $\boldsymbol{0}$. Thus we complete the proof. □

Theorem 1 sheds light on understanding the observation in Tack et al. (2020); Lee et al. (2018); Sun et al. (2022) that ID features tend to form clusters and OOD features reside far away. As shown in Theorem 1, training features will converge to weight vectors as the training epoch goes to infinity. Since ID test samples are drawing from the same distribution as training samples, we expect that ID test features will also cluster in close proximity to weight vectors. In Figure 1 *Left*, we use UMAP(McInnes et al., 2018) to visualize ID features, OOD features, and weight vectors in the penultimate space. In line with our understanding, the ID feature resides in proximity to weight vectors, and OOD features are far away from weight vectors.

## 3.2 OUT-OF-DISTRIBUTION DETECTION

We now leverage our understanding from Theorem 1 and propose an OOD detector.

As ID features tend to cluster near weight vectors, we propose to detect OOD based on feature proximity to weight vectors. Specifically, we quantify the proximity between centered feature $\boldsymbol{h} - \boldsymbol{\mu}_G$ and the weight vector $\boldsymbol{w}_c$ corresponding to the predicted class $c$ as:

$$\texttt{pScore} = \frac{(\boldsymbol{h} - \boldsymbol{\mu}_G) \cdot \boldsymbol{w}_c}{\|\boldsymbol{h} - \boldsymbol{\mu}_G\|_2} \tag{4}$$

Geometrically, our proximity score pScore measures the norm of the projection of $\boldsymbol{w}_c$ onto the centered feature $\boldsymbol{h} - \boldsymbol{\mu}_G$. And the larger pScore is, the closer the feature is to the weight vector. We illustrate in Figure 1 *Middle Upper* how pScore effectively distinguishes between ID and OOD features. Furthermore, we study in Section 4.4 that the same observation also holds under common similarity metrics, cosine similarity, and Euclidean distance. pScore outperforms both as an OOD indicator, which we will further discuss in Section 4.4. Note that pScore of samples within the same predicted class is proportional to the cosine similarity and $\texttt{pScore} = cos(\boldsymbol{w}_c, \boldsymbol{h} - \boldsymbol{\mu}_G)\|\boldsymbol{w}_c\|_2$. When thresholding on pScore, we geometrically select infinite hyper-cones centered at the weight vectors, as illustrated in Figure 1 *Right*.

Essentially, we provide an alternative approach for characterizing ID clustering in comparison to existing literature (Lee et al., 2018; Sun et al., 2022) which builds auxiliary models over the training features. From the weight vector perspective, our clustering score pScore eliminates the need for an auxiliary model, and therefore eliminates the associated computational cost and storage overhead. We further discuss the performance gain of our approach in Section 4.4.

Furthermore, some OOD features may fall within the hyper-cones in Figure 1 *Right* as they are close to the weight vectors under our angular metric pScore. To filter out such OOD samples, we adopt a complementary perspective and leverage the observation in literature (Tack et al., 2020; Sun et al., 2022) that OOD features tend to reside closer to the origin than ID features. As illustrated in Figure 1*Middle Lower*, OOD features tend to have smaller L1 norm than ID features. Geometrically, the complementary perspective allows us to filter out OOD samples near the origin as visualized in Figure 1 *Right*. Combining both perspectives, we design our OOD detection score as

$$\texttt{NCScore} = \alpha\|\boldsymbol{h}\|_1 + \texttt{pScore} = \alpha\|\boldsymbol{h}\|_1 + \frac{(\boldsymbol{h} - \boldsymbol{\mu}_G) \cdot \boldsymbol{w}_c}{\|\boldsymbol{h} - \boldsymbol{\mu}_G\|_2}, \tag{5}$$

where $\alpha$ controls the strength of L1 norm based filtering. We refer readers to Section 4.4 where we conduct a comprehensive ablation study to understand the effectiveness of each component.

By applying a threshold on NCScore, we introduce **N**eural **C**ollapse inspired **OOD** detector ($\texttt{NC} - \texttt{OOD}$), which identifies samples below the threshold as OOD.

## 4 EXPERIMENTS

In this section, we demonstrate the versatility of $\texttt{NC} - \texttt{OOD}$ across OOD Benchmarks on different classification tasks (Sec 4.1 & Sec 4.3), different training objectives (Sec 4.3) and different network architectures(Sec 4.3). This improves over existing methods which focus more on certain classification tasks than others (Sec 4.1). We conduct a comprehensive ablation study to understand the effectiveness of individual components in our detector (Sec 4.4). In the following, we use the area under the receiver operating characteristic curve (AUROC) as our evaluation metric, which is widely used in OOD detection literature. The higher the AUROC value, the better the performance. In addition to the AUROC score, we further report the FPR95 score, the positive rate at $95\%$ true positive rate, of our experiments in Appendix C. For hyperparameter tuning, we follow approaches in Sun et al. (2021); Sun & Li (2022) and we select filter strength from $\alpha = \{0.001, 0.01, 0.1, 1\}$ based on a validation set of Gaussian noise images, which is generated per pixel from $N(0, 1)$. We refer readers to Appendix A for detailed experimental setups.

### 4.1 VERSATILITY ACROSS DIVERSE SCENARIOS

In the following, we examine the performance of $\texttt{NC} - \texttt{OOD}$ against baseline OOD detectors. Particularly, we are interested in evaluating the versatility of OOD detectors on canonical CIFAR-10

and ImageNet OOD benchmarks. Note that the two classification tasks drastically differ in input resolution, number of classes, and classification accuracy. Specifically, the CIFAR-10 classifier we evaluated has a ResNet-18 backbone and achieves an accuracy of 94.21%. The ImageNet classifier has a ResNet-50 backbone and achieves an accuracy of 76.65%. Both classifiers are trained with cross-entropy in this section. Filter strength $\alpha$ is set to 0.01 for the CIFAR-10 benchmark and 0.001 for the ImageNet benchmark based on validation results on Gaussian noise images.

**CIFAR-10 OOD Benchmark Datasets** For the CIFAR-10 OOD benchmark, we consider the standard CIFAR-10 test set with 10,000 images as ID samples. For OOD samples, we evaluate on *six* OOD benchmarks: SVHN, LSUN-crop, LSUN-resize (Yu et al., 2015), iSUN (Xu et al., 2015), Places365 (Zhou et al., 2017), and Texture (Cimpoi et al., 2014). All images are of size $32 \times 32$.

**ImageNet OOD Benchmark Datasets** We consider 50,000 ImageNet validation images in the standard split as ID samples. Following Huang & Li (2021); Sun et al. (2022), we use Places365 (Zhou et al., 2017), iNaturalist (Van Horn et al., 2018), SUN (Xiao et al., 2010), and Texture (Cimpoi et al., 2014) with non-overlapping classes *w.r.t.* ImageNet as OOD samples. All images are of size $224 \times 224$.

**Baselines** We compare our method with 10 baseline methods in Table 1. We observe that some baselines are more focused on CIFAR-10 Benchmark whereas others are more focused on Imagenet Benchmark. Based on the performance, we divide the baselines, besides the vanilla confidence-based MSP (Hendrycks & Gimpel, 2016), into two categories. Specifically, we name baselines focusing more on CIFAR-10 OOD benchmarks as "CIFAR-10 Strong". This includes ODIN (Liang et al., 2018), Energy (Liu et al., 2020), Mahalanobis (Lee et al., 2018), and KNN(Sun et al., 2022). And we refer to baselines particularly competitive on ImageNet OOD benchmarks as "ImageNet Strong", including GradNorm (Huang et al., 2021), React (Sun et al., 2021), and Dice (Sun & Li, 2022). We also consider Dice + React as a baseline, since the best result on ImageNet OOD benchmarks is achieved by combining Dice and ReAct in Sun & Li (2022),

Note that out of the baselines, Mahalanobis and KNN utilize the clustering of ID features and belong to the same school of thought as us. We further compare the performance of two baselines with our standalone clustering score pScore through an ablation study in Section 4.4.

**OOD detection performance** In Table 1, we compare the AUROC score of our methods with baselines. We observe that CIFAR-10 Strong baselines achieve competitive performance on CIFAR-10 benchmarks while performing sub-optimally over ImageNet benchmarks. ImageNet Strong baselines, on the other hand, significantly outperform CIFAR-10 benchmarks on ImageNet OODs yet do not achieve state-of-the-art performance on most CIFAR-10 OOD tasks.

On the contrary, our proposed method concurrently achieves competitive performance across both benchmarks and on average improves AUROC scores. Overall, our experiments provide strong evidence for the efficacy of our proposed detectors in identifying different types of OOD samples across drastically different scenarios.

## 4.2 Effectiveness under Contrastive Learning Scheme

To examine the generalization of our proposed method beyond classifiers trained with cross-entropy loss, we further experiment with a contrastive learning scheme. We consider four baseline methods particularly competitive under contrastive loss: CSI Tack et al. (2020), SSD+ Sehwag et al. (2020), and KNN+ Sun et al. (2022). In Table 2[1], we evaluate NC − OOD, CSI, SSD+, and KNN+ on a CIFAR-10 classifier with ResNet18 backbone trained with supervised contrastive (supcon) loss Khosla et al. (2020). The classifier achieves an accuracy of 94.64%. We set filter strength $\alpha = 1$ based on the validation result on Gaussian noise images.

From Table 2, we observe that under a contrastive training scheme, our proposed detectors on average improve over OOD detection performance of state-of-the-art. In addition, comparing Table 2 with Table 1, we observe that OOD detection performance significantly improves when the classifier is trained with contrastive loss. This is in line with the observation in Sun et al. (2022) that features learnt from contrastive loss are more compact than features learnt from classical cross-entropy loss.

---

[1]CSI result copied from Table 4 in Sun et al. (2022), which does not report performance on CIFAR10-C, CIFAR100, and LUN-resize.

Table 2: Our proposed OOD detectors achieve high AUROC (higher is better) on CIFAR-10 OOD benchmarks when the classifier is trained with contrastive loss. The larger the value, the better. The best result is highlighted in **bold** and the second best result is underlined.

| Method | SVHN | LSUN-C | LSUN-R | iSUN | Place365 | Texture | Avg |
|--------|------|--------|--------|------|----------|---------|-----|
| CSI | 94.69 | 98.84 | NA | 98.01 | 93.04 | 94.87 | NA |
| SSD+ | **99.72** | 98.48 | 95.42 | 95.16 | 95.48 | 97.70 | 96.99 |
| KNN+ | 99.57 | 99.48 | 96.81 | 96.74 | **96.57** | **98.56** | 97.95 |
| NC-OOD | 99.12 | **99.82** | **98.25** | **98.35** | 95.40 | 97.65 | **98.10** |

Table 3: Our proposed OOD detectors achieve competitive performance on DenseNet-101 CIFAR-10 classifier and DenseNet-101 CIFAR-100 classifier. We report the AUROC score averaged over OOD test sets listed in Section 4.1. The larger the value, the better. The best result is highlighted in **bold** and the second best result is underlined.

|  | MSP | ODIN | Energy | Mahalanobis | KNN | GradNorm | ReAct | DICE | ReAct+DICE | NC-OOD |
|--|-----|------|--------|-------------|-----|----------|-------|------|------------|--------|
| CIFAR-10 | 92.46 | 93.71 | 94.57 | 89.15 | **97.20** | 93.96 | 94.62 | 95.24 | 94.98 | 96.89 |
| CIFAR-100 | 74.36 | 84.49 | 81.19 | 82.73 | 84.78 | 81.02 | 78.64 | 86.48 | **86.79** | 86.18 |

## 4.3 EFFECTIVENESS ON DENSENET

So far our evaluation shows that $NC-OOD$ stays competitive on the ResNet backbone across OOD Benchmarks over different classification tasks and classifiers under different training objectives. We now extend our evaluation to DenseNet (Huang et al., 2017). Besides a CIFAR-10 classifier, we consider in addition a CIFAR-100 classifier. The CIFAR-10 classifier achieves a classification accuracy of $94.53\%$, whereas the CIFAR-100 classifier achieves an accuracy of $75.06\%$. We consider the same OOD test sets for CIFAR-10 and CIFAR-100 as in Section 4.1 for CIFAR-10. For the sake of space, we report the AUROC score averaged over test OOD data sets in Table 6 and report the complete results in Appendix D. For both CIFAR-10 and CIFAR-100, the filtering strength $\alpha$ is set to $0.01$. The competitive performance shown in Table 6 further indicates the versatility of our proposed detector across different network architectures and OOD benchmarks.

## 4.4 ABLATION STUDY

In this section, we conduct a comprehensive ablation study to understand the effectiveness of individual components in our $NCScore$ in Section 4.1. Further, we experiment with alternative metrics to gain insight. Note that besides this section, our score function stays the same across OOD datasets, classification tasks, architectures, and training loss throughout the paper.

### 4.4.1 EFFECTIVENESS OF INDIVIDUAL COMPONENTS

In Table 4, we present the performance of OOD detection using $NCscore$ and its standalone components $pScore$ and $L1$ as scoring function on ImageNet OOD benchmarks. While each individual component can distinguish between ID and OOD, we observe across OOD sets that the joint score $NCScore$ largely outperforms both individual components.

Particularly, when comparing Table 4 to Table 1, we observe that detection based on $pScore$ alone, which characterizes ID clustering from the perspective of weight vectors, consistently outperforms existing feature layer methods, $Mahalanobis$(Lee et al., 2018) and $KNN$(Sun et al., 2022), across all OOD sets. Similarly, we observe in Appendix E that, on CIFAR-10 Benchmark, $pScore$ alone achieves an average AUROC of 95.21, outperforming all baselines including $Mahalanobis$ and $KNN$. Recall that $Mahalanobis$ detects OOD based on the Mahalanobis distance between the feature and the mixture of Gaussian distribution learnt from training features; $KNN$ measures OOD-ness based on the k-th nearest neighbor distance to training features. In comparison to $Mahalanobis$, we eliminate the bias in Gaussian assumption, which does not necessary hold as shown in Sun et al. (2022). And in comparison to $KNN$, our method is more robust against outliers in training features. Overall, in addition to the computation and storage benefit in comparison to existing feature layer methods as

Table 4: Effectiveness of components in `NCScore`. AUROC scores are reported (higher is better) on ImageNet OOD Benchmarks. `NCScore` outperforms its components `pScore` and L1 norm.

| Method | iNaturalist | SUN | Places | Texture | Avg |
|---|---|---|---|---|---|
| pScore | 92.65 | 86.52 | 83.75 | 91.86 | 88.69 |
| L1 norm | 90.84 | 87.14 | 82.12 | 88.21 | 87.08 |
| NCScore | 96.94 | 93.46 | 89.91 | 95.65 | 94.15 |

discussed in Section 3.2, our characterization further reduces the bias in assumption as well as the noise in estimation, leading to improved performance.

Furthermore, comparing the performance `NCScore` with `pScore`, we observe that across all OOD datasets in Table 4, filtering on L1 norm improves the performance. The observation validate that L1 norm based filtering can effectively rule out OOD samples falls near weight vectors under metric `pScore` following our intuition in Section 3. Experiments on CIFAR-10 Benchmarks in Appendix E also demonstrate the effectiveness of filtering. However, the enhancement is less significant. Intuitively, the classification of CIFAR-10 is an easier task. Therefore, for a CIFAR-10 classifier, ID features of different classes are better separated whereas ID features of the same class are more compact. And the `pScore` will select much narrower hyper-cones in Figure 1, resulting in a low chance of random OOD falling into the hyper-cones. This matches the observation that `pScore` alone achieves an average AUROC score of $95.21$ in Appendix E, which saturates the state-of-the-art performance. Thus further filtering does not lead to significant improvement.

### 4.4.2 ALTERNATIVES METRICS

**Alternative similarity metrics.** We start by validating that under alternative similarity metrics, ID features also reside closer to weight vectors. Further, we study how to most effectively characterize feature proximity to weight vectors for OOD detection. Specifically, in addition to our proposed `pScore` we consider two standard similarity metrics, cosine similarity and Euclidean distance. For cosine similarity, we evaluate

$$\texttt{cosScore} = \frac{(\boldsymbol{h} - \boldsymbol{\mu}_G) \cdot \boldsymbol{w}_c}{\|\boldsymbol{h} - \boldsymbol{\mu}_G\|_2 \|\boldsymbol{w}_c\|_2}. \tag{6}$$

As for Euclidean distance, we first estimate the scaling factor in Theorem 1 by $\tilde{\lambda}_c = \frac{\|\boldsymbol{\mu}_c - \boldsymbol{\mu}_G\|_2}{\|\boldsymbol{w}_c\|_2}$.

Based on the estimation, we measure the distance between the centered feature $\boldsymbol{h} - \boldsymbol{\mu}_G$ and the scaled weight vector corresponding to the predicted class $c$ as

$$\texttt{distScore} = -\|(\boldsymbol{h} - \boldsymbol{\mu}_G) - \tilde{\lambda}_c \boldsymbol{w}_c\|_2. \tag{7}$$

Same as `pScore`, the larger `cosScore` or `distScore` is, the closer the feature is to the weight vector.

We evaluate in Table 5 OOD detection performance using standalone `pScore`, `cosScore`, and `distScore` as scoring function respectively. The experiments are evaluated with AUROC under the same ImageNet setup as in Section 4.1. We observe in Table 5, that across OOD datasets, all three scores achieve an AUROC score $> 50$, indicating that ID features reside closer to weight vectors compared to OOD under either metric. The same observation also holds on the CIFAR-10 Benchmark, as presented in Appendix E.

Furthermore, we observe that `pScore` outperforms both `cosScore` and `distScore`. And we observe in Appendix E the same pattern on CIFAR-10 Benchmarks. Comparing the performance of `pScore` and `cosScore`, the superior performance of `pScore` implies that ID features corresponding to the classes with larger $\boldsymbol{w}_c$ are less compact. This is inline with the decision rule of the classifier that classes with larger $\boldsymbol{w}_c$ have larger decision regions. As for comparison against Euclidean distance based `distScore`, `pScore` eliminates the need to estimate the scaling factor, which can be error-prone before convergence, potentially leading to performance degradation.

**Alternative filtering norms.** In addition, we ablate on L1, L2, and `Linf` to determine the $p$-norm for filtering in our score function `NCScore`. In Table 6 we compare the performance when filtering `pScore` with different norms. Experiments are on ImageNet Benchmarks under the same setup as Section 4.1. And the selected filtering strength $\alpha$ is 0.001 for all choices of norm. We observe in Table 6, that across OOD datasets, filtering with L1 norm achieves the best OOD detection performance. The same observation also holds on the CIFAR-10 Benchmark in Appendix E.

Table 5: Ablation on similarity scores. AUROC score is reported (higher is better). ID features are closer to weight vectors than OOD features (AUROC > 50) under all metrics. Our proposed `pScore` can better separate ID an OOD features than `distScore` and `cosScore`.

| Method | iNaturalist | SUN | Places | Texture | Avg |
|--------|-------------|-------|--------|---------|-------|
| distScore | 68.96 | 85.78 | 72.31 | 87.07 | 78.53 |
| cosScore | 90.41 | 83.13 | 80.21 | 91.35 | 86.28 |
| pScore | 92.65 | 86.52 | 83.75 | 91.86 | 88.69 |

Table 6: Ablation on filtering norm. AUROC score is reported (higher is better). Filtering with `L1` norm outperform alternative choice of norm.

| Method | iNaturalist | SUN | Places | Texture | Avg |
|--------|-------------|-------|--------|---------|-------|
| Filtering w/ L1 | 96.94 | 93.46 | 89.91 | 95.65 | 94.15 |
| Filtering w/ L2 | 92.83 | 86.81 | 84.02 | 92.00 | 88.92 |
| Filtering w/ Linf | 92.57 | 86.47 | 83.69 | 91.81 | 88.64 |

## 5 RELATED WORK

**OOD Detection** An extensive body of research work has been focused on developing OOD detection algorithms. One line of work is post-hoc and builds upon pre-trained models. For example, Liang et al. (2018); Hendrycks et al. (2019); Liu et al. (2020); Sun et al. (2021); Sun & Li (2022) design OOD score over the output space of a classifier. Meanwhile, Lee et al. (2018); Sun et al. (2022) measure OOD-ness from the perspective of ID clustering in *feature* space. Specifically, Lee et al. (2018) models the feature distribution as multivariate Gaussian and measures OOD based on Mahalanobis distance, whereas Sun et al. (2022) builds upon nearest-neighbor distance. Our work also builds upon the observation that ID features tend to cluster and provides further understanding from the perspective of Neural Collapse. While existing work is more focused are certain classification tasks than others, our proposed OOD detector is tested to be highly versatile.

Another line of work explores the regularization of OOD detection in training. For example, De-Vries & Taylor (2018); Hsu et al. (2020) propose OOD-specific architecture whereas Wei et al. (2022); Huang & Li (2021) design OOD-specific training loss. In particular, Tack et al. (2020) brings attention to representation learning for OOD detection and proposes an OOD-specific contrastive learning scheme. Our work is not restricted to specific training schemes or architecture. Meanwhile, we explore in experiments the benefit of contrastive learning schemes.

**Neural Collapse** Neural Collapse was first observed in Papyan et al. (2020). During Neural Collapse, the penultimate layer features collapse to class means, the class means and the classifier collapses to a simplex equiangular tight framework, and the classifier simplifies to adopt the nearest class-mean decision rule. Further work provides theoretical justification for the emergence of Neural Collapse(Zhu et al., 2021; Zhou et al., 2022; Han et al., 2021; Mixon et al., 2020). In addition, Zhu et al. (2021) derives an efficient training algorithm drawing inspiration from Neural Collapse. To the best of our knowledge, we are the first to leverage insights from Neural Collapse to distinguish between ID and OOD samples and henceforth to develop a practical OOD detection algorithm.

## 6 CONCLUSION

This work takes inspiration from Neural Collapse to propose a novel OOD detector. Specifically, we study the phenomenon that ID features tend to form clusters whereas OOD features reside far away. We demonstrate from Neural Collapse that ID features tend to reside near weight vectors. We combine our understanding with the observation that OOD features tend to reside closer to the origin to propose an OOD detector. Experiments show that our method can achieve state-of-the-art OOD detection performance across diverse setups, improving upon the generalizability of existing work. We hope our work can inspire future work to explore the duality between features and weight vectors for OOD detection and other research problems such as calibration and adversarial robustness.

## 7 REPRODUCIBILITY STATEMENT

For our algorithm, we provide code in the supplementary material. For the theoretical result, we include the complete proof and assumption in Section 3. For datasets, we use common datasets and data processing steps can be found in our code.

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

## A    IMPLEMENTATION DETAILS

### A.1    CIFAR-10

**ResNet-18 w/ Cross Entropy Loss** For experiments on CIFAR-10 Benchmark presented in Fig. 1*Left, Middle* and Table 1, we evaluate on a CIFAR-10 classifier of ResNet-18 backbone trained with cross-entropy loss. The classifier is trained for 100 epochs, with the initial learning rate 0.1 decaying to 0.01, 0.001, and 0.0001 at epochs 50, 75, and 90 respectively.

**ResNet-18 w/ Contrastive Loss** For Table 2, we experiment with a CIFAR-10 classifier of the ResNet-18 backbone trained with supcon loss. Following Khosla et al. (2020), the model is trained for 500 epochs with batch size 1034. The temperature is set to 0.1. The cosine learning rate starts at 0.5 Loshchilov & Hutter (2016)is used.

**DenseNet-101 w/ Cross Entropy Loss**

For experiments on CIFAR-10 Benchmark presented in Table 3, we evaluate a CIFAR-10 classifier of DenseNet-101 backbone. The classifier is trained following the setup in Huang et al. (2017) with depth $L = 100$ and growth rate $k = 12$.

### A.2    CIFAR-100

**DenseNet-101 w/ Cross Entropy Loss**

For experiments on the CIFAR-100 Benchmark presented in Table 3, we evaluate a CIFAR-100 classifier of the DenseNet-101 backbone. The classifier is trained following the setup in Huang et al. (2017) with depth $L = 100$ and growth rate $k = 12$.

### A.3    IMAGENET

**ResNet-50 w/ Cross-Entropy Loss** For evaluation on ImageNet Benchmark in Table 1, we use the default ResNet-50 model trained with cross-entropy loss provided by Pytorch. Training recipe can be found at `https://pytorch.org/blog/how-to-train-state-of-the-art-models-using-torchvision-latest-primitives/`

## B    BASELINE METHODS

We provide an overview of our baseline methods in this session. We follow our notation in Section 3. In the following, a lower detection score indicates OOD-ness.

**MSP** Hendrycks & Gimpel (2016) proposes to detect OOD based on the maximum softmax probability. Given the penultimate feature $\boldsymbol{h}$ for a given test sample $\boldsymbol{x}$, the detection score of MSP can be represented as:

$$\frac{\exp\left(\boldsymbol{w}_c^T \boldsymbol{h} + b_c\right)}{\sum_{c' \in \mathcal{C}} \exp\left(\boldsymbol{w}_{c'}^T \boldsymbol{h} + b_{c'}\right)}, \tag{8}$$

where $c$ is the predicted class for $\boldsymbol{x}$.

**ODIN** Liang et al. (2018) proposes to amplify ID the OOD separation on top of MSP through temperature scaling and adversarial perturbation. Given a sample $\boldsymbol{x}$, ODIN constructs a noisy sample $\boldsymbol{x}'$ from $\boldsymbol{x}$. Denote the penultimate feature of the noisy sample $\boldsymbol{x}'$ as $\boldsymbol{h}'$, ODIN assigns OOD score following:

$$\frac{\exp\left((\boldsymbol{w}_c^T \boldsymbol{h}' + b_c)/T\right)}{\sum_{c' \in \mathcal{C}} \exp\left((\boldsymbol{w}_c'^T \boldsymbol{h}' + b_{c'})/T\right)}, \tag{9}$$

where $c$ is the predicted class for the perturbed sample and $T$ is the temperature. In our implementation, we set the noise magnitude as 0.0014 and the temperature as 1000.

**Energy** Liu et al. (2020) designs an energy-based score function over the logit output. Given a test sample $\boldsymbol{x}$ as well as its penultimate layer feature $\boldsymbol{h}$, the energy based detection score can be represented as:

$$-\log \sum_{c' \in \mathcal{C}} \exp\left(\boldsymbol{w}_{c'}^T \boldsymbol{h} + b_{c'}\right). \tag{10}$$

**ReAct** Sun et al. (2021) builds upon the energy score proposed in Liu et al. (2020) and regularizes the score by truncating the penultimate layer estimation. We set the truncation threshold at 90 percentile in our experiments.

**Dice** Sun & Li (2022) also builds upon the energy score proposed in Liu et al. (2020). Leveraging the observation that units and weights are used sparsely in ID inference, Sun & Li (2022) proposes to select and compute the energy score over a selected subset of weights based on their importance. We set a threshold at 90 percentile for CIFAR experiments and 70 percentile for ImageNet experiments following Sun & Li (2022).

**Mahalanobis** On the feature space, Lee et al. (2018) models the ID feature distribution as multivariate Gaussian and designs a Mahalanobis distance-based score:

$$\max_{c} -(\boldsymbol{e_x} - \hat{\boldsymbol{\mu}}_c)^T \hat{\Sigma}^{-1} (\boldsymbol{e_x} - \hat{\boldsymbol{\mu}}_c), \tag{11}$$

where $\boldsymbol{e_x}$ is the feature embedding of $\boldsymbol{x}$ in a specific layer, $\hat{\mu}_c$ is the feature mean for class $c$ estimated on the training set, and $\hat{\Sigma}$ is the covariance matrix estimated over all classes on the training set.

On top of the basic score, Lee et al. (2018) also proposes two techniques to enhance the OOD detection performance. The first is to inject noise to samples. The second is to learn a logistic regressor to combine scores across layers. We tune the noise magnitude and learn the logistic regressor on an adversarial constructed OOD dataset. The selected noise magnitude is 0.005 in both our ResNet and DenseNet experiments.

**CSI** Tack et al. (2020) proposes an OOD specific contrastive learning algorithm. In addition, Tack et al. (2020) defines detection functions on top of the learned representation, combining two aspects: (1) the cosine similarity between the test sample embedding to the nearest training sample embedding and (2) the norm of the test sample embedding.

**SSD** Similar to Lee et al. (2018), Sehwag et al. (2020) design a Mahalanobis-based score under a representation learning scheme. In specific, Sehwag et al. (2020) proposes a cluster-conditioned score:

$$\max_{m} -(\boldsymbol{e_x}/|\boldsymbol{e_x}| - \hat{\mu}_m)^T \hat{\Sigma}_m^{-1} (\boldsymbol{e_x}/|\boldsymbol{e_x}| - \hat{\mu}_m), \tag{12}$$

where $\boldsymbol{e_x}/|\boldsymbol{e_x}|$ is the normalized feature embedding of $\boldsymbol{x}$ and $m$ corresponds to the cluster constructed from the training statistics.

**KNN** Chen et al. (2020) proposes to detect OOD based on the k-th nearest neighbor distance between the normalized embedding of the test sample $\boldsymbol{z_x}/|\boldsymbol{z_x}|$ and the normalized training embeddings on the penultimate space. Chen et al. (2020) also observes that contrastive learning helps in improving OOD detection effectiveness.

**GradNorm** Huang et al. (2021) extracts information from the gradient space to detect OOD samples. Specifically, Huang et al. (2021) defines the OOD score function as the L1 norm of the gradient of the weight matrix with respect to the KL divergence between the softmax prediction for $\boldsymbol{x}$ and the uniform distribution.

$$\|\frac{\partial D_{KL}(\boldsymbol{u}\|softmax f(\boldsymbol{x}))}{\partial \boldsymbol{W}}\|_1. \tag{13}$$

## C  PERFORMANCE UNDER FPR95

In addition to the AUROC score reported in the main paper, we also compare the performance of our $\texttt{NC} - \texttt{OOD}$ with baselines under FPR95, the false positive rate of OOD samples when the tur positive rate of ID samples is at $95\%$. Specifically, we report Table 7 corresponding to experiments in Table 1

Table 7: **Our proposed OOD detector achieve low FPR95 score across CIFAR-10 and ImageNet OOD benchmark.** The smaller the value, the better. The experiments are under the same setup as Table 1 in Section 4.1

| | Method | CIFAR-10 OOD Benchmark | | | | | | | ImageNet OOD Benchmark | | | | |
|---|---|---|---|---|---|---|---|---|---|---|---|---|---|
| | | SVHN | LSUN-C | LSUN-R | iSUN | Place365 | Texture | AVG | iNaturalist | SUN | Places | Texture | Avg |
| CIFAR-10 Strong | MSP | 59.51 | 45.27 | 51.96 | 54.57 | 62.55 | 66.49 | 56.72 | 54.99 | 70.83 | 73.99 | 68.00 | 66.95 |
| | ODIN | 61.71 | 8.71 | 12.54 | 15.09 | 41.45 | 52.62 | 32.02 | 47.66 | 60.15 | 67.90 | 50.23 | 56.48 |
| | Energy | 53.96 | 10.19 | 23.47 | 27.52 | 42.80 | 55.23 | 35.53 | 55.72 | 59.26 | 64.92 | 53.72 | 58.41 |
| | Maha | 16.77 | 27.67 | 6.68 | 7.56 | 85.87 | 35.21 | 30.01 | 97.00 | 98.50 | 98.40 | 55.80 | 87.43 |
| | KNN | 27.85 | 18.5 | 22.68 | 24.67 | 44.56 | 37.57 | 29.31 | 59.00 | 68.82 | 76.28 | 11.77 | 53.97 |
| ImageNet Strong | GradNorm | 82.44 | 22.55 | 48.94 | 54.14 | 69.26 | 69.38 | 57.79 | 26.93 | 37.22 | 48.67 | 32.59 | 36.35 |
| | ReAct | 46.50 | 23.03 | 19.11 | 22.02 | 47.23 | 48.12 | 34.34 | 20.38 | 24.20 | 33.85 | 47.30 | 31.42 |
| | DICE | 65.04 | 3.94 | 29.79 | 34.85 | 49.83 | 59.36 | 40.47 | 25.63 | 35.15 | 46.49 | 31.72 | 34.75 |
| | DICE + ReAct | 47.36 | 9.90 | 18.50 | 22.49 | 61.21 | 42.30 | 33.63 | 18.64 | 25.45 | 36.86 | 28.07 | **27.25** |
| | NC-OOD | 22.36 | 10.76 | 20.42 | 23.42 | 48.04 | 30.64 | **25.94** | 16.64 | 30.38 | 42.56 | 19.52 | 27.28 |

Table 8: Our proposed OOD detectors achieve low FPR95 (higher is better) on CIFAR-10 OOD benchmarks when the classifier is trained with contrastive loss. The smaller the value, the better. The experiments are under the same setup as Table 2 in Section 4.2.

| Method | SVHN | LSUN-C | LSUN-R | iSUN | Place365 | Texture | Avg |
|---|---|---|---|---|---|---|---|
| CSI | 37.38 | 5.88 | NA | 10.36 | 38.31 | 28.85 | NA |
| SSD+ | 1.35 | 6.09 | 31.25 | 33.60 | 26.09 | 12.98 | 18.56 |
| KNN+ | 2.20 | 1.78 | 19.13 | 20.06 | 18.38 | 8.09 | 11.61 |
| NC-OOD | 4.50 | 0.48 | 8.87 | 8.47 | 23.98 | 12.74 | **9.84** |

as well as Table 8 corresponding to Table 2. The results in FPR95 further validate the effectiveness of our NC − OOD across diverse scenarios.

# D   COMPLETE RESULTS ON DENSENET

In addition to Table 3, we reports the performance of our NC − OOD and baselines under AUROC and FPR95 across all OOD datasets in TableD and TableD.

# E   ABLATION STUDY ON CIFAR-10 BENCHMARKS

In Table 11, Table 12, and Table 6, we report our ablation study over CIFAR-10 OOD Benchmarks. Note that in Table 6, the filtering strength $\alpha$ is set to $0.01, 0.001, 0.001$ for L1 norm, L2 norm, and Linf norm respectively.

| Method | SVHN | | LSUN_crop | | LSUN_resized | | iSUN | | Place365 | | Texture | | AVG | |
|---|---|---|---|---|---|---|---|---|---|---|---|---|---|---|
| | FPR ↓ | AUROC ↑ | FPR ↓ | AUROC ↑ | FPR ↓ | AUROC ↑ | FPR ↓ | AUROC ↑ | FPR ↓ | AUROC ↑ | FPR ↓ | AUROC ↑ | FPR ↓ | AUROC ↑ |
| MSP | 47.24 | 93.48 | 33.57 | 95.54 | 42.10 | 94.51 | 42.31 | 94.52 | 63.02 | 88.57 | 64.15 | 88.15 | 48.73 | 92.46 |
| ODIN | 25.29 | 94.57 | 4.70 | 98.86 | 3.09 | 99.02 | 3.98 | 98.90 | 52.85 | 88.55 | 57.50 | 82.38 | 24.57 | 93.71 |
| Energy | 40.61 | 93.99 | 3.81 | 99.15 | 9.28 | 98.12 | 10.07 | 98.07 | 39.40 | 91.64 | 56.12 | 86.43 | 26.55 | 94.57 |
| Mahalanobis | 6.42 | 98.31 | 56.55 | 86.96 | 9.14 | 97.09 | 9.78 | 97.25 | 85.14 | 63.15 | 21.51 | 92.15 | 31.42 | 89.15 |
| KNN | 3.96 | 99.29 | 6.91 | 98.75 | 9.38 | 98.24 | 9.54 | 98.27 | 39.96 | 92.24 | 19.52 | 96.38 | 14.88 | 97.20 |
| GradNorm | 25.15 | 93.93 | 0.41 | 99.85 | 9.51 | 98.08 | 10.41 | 97.97 | 41.86 | 90.13 | 43.76 | 83.80 | 21.85 | 93.96 |
| ReAct | 38.81 | 94.19 | 3.81 | 99.15 | 8.77 | 98.19 | 9.54 | 98.11 | 39.37 | 91.66 | 56.33 | 86.41 | 26.10 | 94.62 |
| Dice | 25.99 | 95.90 | 0.26 | 99.92 | 3.91 | 99.20 | 4.36 | 99.14 | 48.59 | 89.13 | 41.90 | 88.18 | 20.84 | 95.24 |
| ReAct+Dice | 27.78 | 94.96 | 0.38 | 99.90 | 4.22 | 99.06 | 5.18 | 98.99 | 45.29 | 90.02 | 45.99 | 86.96 | 21.47 | 94.98 |
| NC-OOD | 5.33 | 98.77 | 3.24 | 99.34 | 8.18 | 98.33 | 8.77 | 98.21 | 44.57 | 90.68 | 22.73 | 96.02 | 15.47 | 96.89 |

Table 9: OOD detection performance on a CIFAR-10 classifier with backbone DenseNet-101. The experiments correspond to Table 3 in Section 4.3

| Method | SVHN | | LSUN_crop | | LSUN_resized | | iSUN | | Place365 | | Texture | | AVG | |
|---|---|---|---|---|---|---|---|---|---|---|---|---|---|---|
| | FPR ↓ | AUROC ↑ | FPR ↓ | AUROC ↑ | FPR ↓ | AUROC ↑ | FPR ↓ | AUROC ↑ | FPR ↓ | AUROC ↑ | FPR ↓ | AUROC ↑ | FPR ↓ | AUROC ↑ |
| MSP | 81.70 | 75.40 | 60.49 | 85.60 | 85.24 | 69.18 | 85.99 | 70.17 | 82.55 | 74.31 | 84.79 | 71.48 | 80.13 | 74.36 |
| ODIN | 41.35 | 92.65 | 10.54 | 97.93 | 65.22 | 84.22 | 67.05 | 83.84 | 82.32 | 76.84 | 82.34 | 71.38 | 58.14 | 84.49 |
| Energy | 87.46, | 81.85 | 14.72 | 97.43 | 70.65 | 80.14 | 74.54 | 78.95 | 79.20 | 77.72 | 84.15 | 71.03 | 68.45 | 81.19 |
| Mahalanobis | 22.44 | 95.67 | 68.90 | 86.30 | 23.07 | 94.20 | 31.38 | 93.21 | 92.66 | 61.39 | 62.39 | 79.39 | 50.14 | 82.73 |
| KNN | 14.66 | 97.06 | 28.33 | 92.71 | 68.99 | 75.60 | 62.49 | 79.05 | 85.35 | 70.17 | 24.34 | 94.11 | 47.36 | 84.78 |
| GradNorm | 60.81 | 87.74 | 0.65 | 99.78 | 82.20 | 75.48 | 78.68 | 78.14 | 83.18 | 73.02 | 65.19 | 71.93 | 61.79 | 81.02 |
| ReAct | 94.42 | 73.37 | 54.12 | 86.69 | 63.12 | 87.96 | 70.33 | 85.23 | 88.90 | 66.89 | 87.84 | 71.69 | 76.46 | 78.64 |
| Dice | 59.94 | 88.20 | 0.91 | 99.74 | 54.92 | 88.24 | 52.41 | 88.52 | 81.39 | 77.07 | 61.44 | 77.13 | 51.83 | 86.48 |
| ReAct+Dice | 52.34 | 88.85 | 4.43 | 99.08 | 43.67 | 92.26 | 35.66 | 93.46 | 92.87 | 60.02 | 41.01 | 87.05 | 76.45 | 86.79 |
| NC-OOD | 65.55 | 87.63 | 28.88 | 94.37 | 69.95 | 86.32 | 69.47 | 86.99 | 83.04 | 73.30 | 58.85 | 88.49 | 62.62 | 86.18 |

Table 10: OOD detection performance on a CIFAR-100 classifier with backbone DenseNet-101. The experiments correspond to Table 3 in Section 4.3

Table 11: Ablation on similarity scores over CIFAR-10 OOD Benchmarks. AUROC score is reported (higher is better). ID features are closer to weight vectors than OOD features (AUROC > 50) under all metrics. Our proposed `pScore` can better separate ID an OOD features than `distScore` and `cosScore`.

| Method | SVHN | LSUN-C | LSUN-R | iSUN | Place365 | Texture | Avg |
|---|---|---|---|---|---|---|---|
| distScore | 94.88 | 87.53 | 90.91 | 89.90 | 82.02 | 91.06 | 89.38 |
| cosScore | 96.70 | 97.33 | 95.77 | 95.13 | 89.92 | 94.32 | 94.86 |
| pScore | 96.50 | 97.56 | 96.10 | 95.49 | 91.17 | 94.45 | 95.21 |

Table 12: Ablation on additional filtering over CIFAR-10 OOD Benchmarks. AUROC score is reported (higher is better). `NCScore` outperforms both its individual component `pScore` and `L1` norm.

| Method | SVHN | LSUN-C | LSUN-R | iSUN | Place365 | Texture | Avg |
|---|---|---|---|---|---|---|---|
| pScore | 96.50 | 97.56 | 96.10 | 95.49 | 91.17 | 94.45 | 95.21 |
| L1 norm | 71.05 | 94.93 | 83.94 | 82.17 | 63.82 | 76.61 | 78.75 |
| NCScore | 96.17 | 98.15 | 96.58 | 96.04 | 90.48 | 94.75 | 95.36 |

Table 13: Ablation on filtering norm over CIFAR-10 OOD Benchmarks. AUROC score is reported (higher is better). Filtering with `L1` norm outperform alternative choice of norm.

| Method | SVHN | LSUN-C | LSUN-R | iSUN | Place365 | Texture | Avg |
|---|---|---|---|---|---|---|---|
| Filtering w/ L1 | 96.17 | 98.15 | 96.58 | 96.04 | 90.48 | 94.75 | 95.36 |
| Filtering w/ L2 | 96.49 | 97.56 | 96.11 | 95.50 | 91.17 | 94.45 | 95.21 |
| Filtering w/ Linf | 96.50 | 97.56 | 96.10 | 95.49 | 91.17 | 94.45 | 95.21 |

