# OpenReview forum: "Detecting Out-of-distribution with Insights from Neural Collapse"
_ICLR.cc/2024/Conference — Submitted to ICLR 2024_

### Official Review · Reviewer_cGpn · 2023-10-25

**Soundness:** 3 good
**Presentation:** 2 fair
**Contribution:** 2 fair
**Rating:** 5
**Confidence:** 4

**Summary:**

This paper studies out-of-distribution (OOD) detection from a neural collapse (NC) view. Based on the four characteristics of NC, the authors theoretically justify that the training in-domain (ID) features would converge to the class vectors of linear classifier. Accordingly, the method NC-OOD is proposed to quantify the proximity between the feature vector and class vector. The empirical results on ImageNet and CIFAR-10 benchmarks also support the effectiveness of NC-OOD.

**Strengths:**

1. The motivation is sound. The view of neural collapse is novel and interesting, which characterizes recent observations regarding ID feature clustering.
2. The proposed NC-OOD is a simple method for OOD detection.

**Weaknesses:**

1. The proposed NC-OOD is an alternative method for many recent approaches, such as KNN. While the authors claim that efficiency is the advantage of the NC-OOD, no empirical results are given to support this claim. Besides, the improvements of NC-OOD with contrastive learning/DenseNet-101 are also marginal.

2. In Eq. (5), the calculation of NCScore is based on the class vector $w_c$. However, in the test phase of OOD detection, the corresponding label for the test sample is unknown. It is unclear how the method calculates the NCScore in such scenarios. Besides, since OOD datasets often contain classes beyond the scope of training classes, how to find a suitable class vector to determine proximity can be a crucial issue in the final deployment.

3. The effectiveness of NC-OOD heavily relies on the assumption of neural collapse. For a better understanding of this approach, it would be beneficial for the authors to explore the connection between NC scores and NC-OOD performance, especially considering the existing metrics for NC quantification, e.g., class-distance normalized variance (CDNV) [1].

4. Many descriptions are missing. Specifically, 1) The standard OOD detection formula is missing. The authors should give a detailed formula to explain how they determine whether a test sample is OOD or not. 2)  In NC2, the meaning of $\delta_{c_c’}$ is unclear.

5. From Table 4, it can be observed that the L1 norm plays a significant role in the final performance of NCScore. This raises questions about how the proximity measure as a standalone could outperform the existing clustering-based method.

[1] Tomer Galanti, András György, Marcus Hutter. On the Role of Neural Collapse in Transfer Learning. ICLR 2022.

**Questions:**

See Weaknesses.

---

### Official Review · Reviewer_GX46 · 2023-10-29

**Soundness:** 2 fair
**Presentation:** 3 good
**Contribution:** 2 fair
**Rating:** 5
**Confidence:** 3

**Summary:**

Based on the observation of Neural Collapse that the ID features tend to cluster and the difference between ID feature and the global mean converges to a multiplication of its classification head, the paper proposes an OOD detection method via designing a function of feature $h$, feature global mean $\mu_G$, and classification heads $w_c$.
The paper further shows the proposed method has comparable OOD detection performance compared with SOTA methods, and outperforms baselines when the classifier is pretrained with contrastive loss.

**Strengths:**

(1) The method is well motivated by Neural Collapse, and the observation is promising for a good OOD detection performance.

(2) The writing is clear and easy to follow, and the method is straightforward.

(3) It's interesting to see the proposed method is especially effective when the classifier is pretrained with contrastive loss.

**Weaknesses:**

(1) The design philosophy of pScore is unclear. Based on feature, feature global mean, and classification heads, one can design multiple score functions such as simply using the distance between feature and a multiplication of classification heads which is pretty similar to the Mahalanobis score (the paper also briefly discusses the similarity between the proposed method and Mahalanobis). Thus it's not clear why choosing cosine similarity works better than simply using $L_2$ distance.

(2) In NCScore (equation 5), when $h$ is very large, it should be regarded as OOD. However, NCScore will be large when $h$ is very large.

(3) Table 1 shows Mahalanobis works better than NC-OOD on LSUN-R and iSUN. It's unclear under what conditions Mahalanobis outperforms NC-OOD, or when a cosine similarity-based score is superior to a distance-based score. In other words, NC suggests that all features will converge, but the process of this convergence is unclear.

**Questions:**

Why the proposed NCScore is $\alpha\|h\|_1+\frac{(h-\mu_G)w_c}{\|h-\mu_G\|_2}$ rather than $\alpha\|h-\mu_G\|_1+\frac{(h-\mu_G)w_c}{\|h-\mu_G\|_2}$? The question is raised due to the later has a more symmetric score.

---

### Official Review · Reviewer_ayHR · 2023-10-31

**Soundness:** 2 fair
**Presentation:** 2 fair
**Contribution:** 2 fair
**Rating:** 3
**Confidence:** 4

**Summary:**

The passage introduces a versatile Out-of-Distribution (OOD) detector named Neural Collapse inspired OOD detector (NC-OOD), aiming to improve the generalizability of existing OOD detectors for safer AI deployment. It builds on the observation that in-distribution (ID) features usually form clusters, while OOD features are more scattered. Leveraging the Neural Collapse phenomenon, where ID features cluster near weight vectors, the method proposes to detect OOD samples based on their proximity to these weight vectors. Additionally, it utilizes the fact that OOD features tend to be closer to the origin than ID features to enhance detection. Extensive testing demonstrates that this approach significantly boosts generalizability and consistently outperforms existing methods across various benchmarks, classification tasks, loss functions, and model architectures.

**Strengths:**

1. The introduction of Neural Collapse for OOD detection is interesting.
2. The paper is well-written and easy to understand.

**Weaknesses:**

1. In this paper, the authors propose NC-OOD based on neural collapse theory, which analyses the relationship between penultimate layer features and the weight vectors of the classification head in the Terminal Phase of Training (TPT, i.e., train the network with infinite steps and the training loss is trained towards zero). The authors design a theoretical explanation of the proposed OOD score NCScore based on this ideal situation. However, we know that when the network reaches TPT, it itself is in a serious overfitting state.

On the one hand, it may be an unreasonable assumption to think that "ID test samples are drawing from the same distribution as training samples" (the last paragraph on page 4). Because at this time the decision boundary of the classifier is "exactly" determined by the training samples and cannot generalize well on the ID test samples that are "same distributed" with training data.

On the other hand, in reality, no model developer will allow their network to be in TPT, whereas OOD detection is an area that studies the security of models during the deployment phase. We don't know how big the gap is between a normally trained model and TPT, and therefore we don't know how big the gap is between the theoretical analysis of the paper and its actual use for OOD detection.

Therefore, I suggest that the author give detailed explanations, experiments or theoretical analysis to prove that Neural Collapse is still reasonable when the model does not reach TPT, and try to give the values of relative errors of Lemma NC1-NC4. I think Fig. 1 - Left can only illustrate the clustering properties of the classifier, which has been widely revealed in previous studies, but cannot prove the correctness of the Neural Collapse theory in general models (non-TPT).

2. The proposed pScore measures the cosine distance between the centered feature and the category weight, and the L1-norm score calculates the activation strength of the penultimate layer feature. For pScore, there have been works such as KNN [cite] and Mahalanobis distance [cite] to detect OOD samples based on cluster centers. The authors should discuss what are the essential differences between the proposed method and these baseline methods and what are the advantages. For L1-Norm, I wonder why OOD samples have smaller L1-Norm score than ID samples based on the author's Neural Collapse theory. In Fig. 1 - Right, if the OOD samples are distributed in the hypercone directly above, will they have larger L1-Norm values than the ID samples? In addition, using L1-norm of the penultimate layer features as the basis for OOD detection has been studied in the previous work GradNorm [1] (Eqn. 9). I suggest the authors to further explore this issue and provide comparative experiments of KNN + L1-Norm or Mahalanobis distance + L1-Norm.

 3.  It would be beneficial for the author to include near-ood experiments [2] to enhance the comprehensiveness of the study.

4. A number of recent related works have not been compared in this study, including ViM [3], ASH [4], NPOS [5], and CIDER [6].




[1] Rui Huang, Andrew Geng, and Yixuan Li. On the importance of gradients for detecting distributional shifts in the wild. Advances in Neural Information Processing Systems, 34:677–689, 2021.

[2] Fort, Stanislav, Jie Ren, and Balaji Lakshminarayanan. "Exploring the limits of out-of-distribution detection." Advances in Neural Information Processing Systems 34 (2021): 7068-7081.

[3] Wang, Haoqi, et al. "Vim: Out-of-distribution with virtual-logit matching." Proceedings of the IEEE/CVF conference on computer vision and pattern recognition. 2022.

[4] Djurisic, Andrija, et al. "Extremely Simple Activation Shaping for Out-of-Distribution Detection." The Eleventh International Conference on Learning Representations. 2022.

[5] Tao, Leitian, et al. "Non-parametric Outlier Synthesis." The Eleventh International Conference on Learning Representations. 2022.

[6] Ming, Yifei, et al. "How to Exploit Hyperspherical Embeddings for Out-of-Distribution Detection?." The Eleventh International Conference on Learning Representations. 2022.

**Questions:**

see weakness

---

### Official Review · Reviewer_vw3f · 2023-11-01

**Soundness:** 2 fair
**Presentation:** 3 good
**Contribution:** 2 fair
**Rating:** 5
**Confidence:** 4

**Summary:**

The paper proposes a new out-of-distribution (OOD) detection score in the multi-class classification setting. The score is derived based on the combination of the feature norm and the distance to the weight vector of the predicted ID class. The intuition is that ID samples tend to reside close to weight vectors while OOD samples are far away after training for long period of time. Experiments on CIFAR and ImageNet demonstrate competitive performance of the proposed OOD score.

**Strengths:**

- The overall organization of the paper is clear and easy to follow.
- The proposed approach is quite straightforward and easy to understand.

**Weaknesses:**

The main hypothesis of the paper is not convincingly justified. While the neural collapse phenomenon suggests that ID samples align with their respective ID class weight vectors, this doesn't directly imply that OOD samples will deviate from these vectors. Many established OOD scores, like KNN+ and Mahalanobis, operate on the same core idea: after training, OOD samples tend to be far away from ID clusters in the feature space. Given that one can use ID samples for the mean feature vector in KNN+, the novelty of the proposed method in contrast to prior works seems limited.

Earlier works have pointed out that "OOD features often lie closer to the origin" [1]. Thus, it is expected that adding feature norm to the OOD score can improve the performance. It remains underexplored in principle and theory the specific conditions and assumptions under which OOD features have smaller norms.

The approach's computational overhead is notable. Given that neural collapse usually implies the "training epoch approaching infinity", the method may be computationally intensive. Encouraging ID samples to align directly with their ID prototypes during training might be more efficient such as [2]. Such comparisons are missing in the paper.

**Questions:**

Method:
- I wonder how neural collapse implies that OOD feature will be far from ID weight vectors? Note that training is only conducted on ID samples.
- I do not quite understand why OOD feature norms tend to be small. Can authors provide a more reasonable justification (preferably a more principled or theoretical justification)?

Experiments:

- If neural collapse is desirable, instead of training for a long period of time, encouraging ID samples to align directly with their ID prototypes during training might be more efficient [2]. Is the proposed OOD detection suitable for prototype-based contrastive losses [2] as well?

[1] Tack et al., CSI: Novelty detection via contrastive learning on distributionally shifted instances, NeurIPS 2020

[2] Ming et al., How to Exploit hyperspherical embeddings for out-of-distribution detection?, ICLR 2023

---

### Meta-Review · Area_Chair_Bo5t · 2023-12-08

**Metareview:**

This paper investigates the out-of-distribution detection problem from the neural collapse view. Although the idea is interesting and deserves attention from the community, the main insight of the paper is not convincingly justified, as concerned by most of the reviewers. Besides, some important related references and the comparisons with them are missing. The authors provided no response. The ACs agree with the reviewers and thus recommend reject for this paper.

**Justification For Why Not Higher Score:**

Important references and comparisons are missing. The proposed idea is not well supported to be convincing.

**Justification For Why Not Lower Score:**

N/A

---

### Decision · Program_Chairs · 2024-01-16

Reject